medical computing/computational biology

three-dimensional ultrasound, craniofacial abnormalities, image segmentation

**Author for correspondence:**
C. C. Lees
e-mail: christoph.lees@nhs.net

# Developing and testing an algorithm for automatic segmentation of the fetal face from three-dimensional ultrasound images

A. E. Clark[1,2], B. Biffi[2], R. Sivera[2], A. Dall'Asta[1,2,3], L. Fessey[2], T.-L. Wong[1], G. Paramasivam[1,2], D. Dunaway[4,5], S. Schievano[4,5] and C. C. Lees[1,6]

[1]Queen Charlotte's and Chelsea Hospital, Imperial Healthcare NHS Trust, London, UK
[2]Imperial College London, London, UK
[3]Department of Medicine and Surgery, Obstetrics and Gynaecology Unit, University of Parma, Italy
[4]University College London GOS Institute of Child Health, London, UK
[5]Great Ormond Street Hospital for Children, London, UK
[6]Institute of Reproductive and Developmental Biology, Department of Metabolism, Digestion and Reproduction, Imperial College London, London, UK

AEC, 0000-0002-9556-4727; CCL, 0000-0002-2104-5561

Fetal craniofacial abnormalities are challenging to detect and diagnose on prenatal ultrasound (US). Image segmentation and computer analysis of three-dimensional US volumes of the fetal face may provide an objective measure to quantify fetal facial features and identify abnormalities. We have developed and tested an atlas-based partially automated facial segmentation algorithm; however, the volumes require additional manual segmentation (MS), which is time and labour intensive and may preclude this method from clinical adoption. These manually refined segmentations can then be used as a reference (atlas) by the partially automated segmentation algorithm to improve algorithmic performance with the aim of eliminating the need for manual refinement and developing a fully automated system. This study assesses the inter- and intra-operator variability of MS and tests an optimized version of our automatic segmentation (AS) algorithm. The manual refinements of 15 fetal faces performed by three operators and repeated by one operator were assessed by Dice score, average symmetrical surface distance and volume difference. The performance of the partially automatic algorithm with difference size atlases was evaluated by Dice score and computational time. Assessment

of the manual refinements showed low inter- and intra-operator variability demonstrating its suitability for optimizing the AS algorithm. The algorithm showed improved performance following an increase in the atlas size in turn reducing the need for manual refinement.

# 1. Introduction

Fetal craniofacial abnormalities provide valuable clues in diagnosing genetic conditions and syndromes. However, many of these phenotypical features can be subtle and, therefore, challenging to identify on prenatal two-dimensional (2D) ultrasound (US) even for the most expert of investigators. The use of three-dimensional (3D) US scans complement 2D findings and can provide additional information, both aiding the identification of abnormalities *in utero* as well as providing additional diagnostic clues, thus facilitating targeted genetic testing [1–5].

Traditionally, analysis of 3D US scans is subjective, operator dependent and heavily reliant on the breadth of the operator's experience and with rarer conditions evading detection as their features are not universally well known. Computerized analysis of 3D US volumes could provide an objective method to characterize fetal facial morphology [6], which may assist in identifying conditions with unknown genetic factors but well-characterized craniofacial features and improve the diagnosis of phenotypically heterogenous conditions.

In order to achieve this, a surface representation or map of the fetal face is extracted from the 3D volume via image segmentation, a process to divide an image or volume into non-overlapping separate parts or 'segments' to facilitate volumetric image processing and analysis [7]. In previous work by this group, manual segmentation (MS) was used to delineate the fetal face surfaces meshes from 20 3D US volumes [6]. This is a particularly time-consuming process which precludes it from everyday clinical use.

The use of various methods for the automatic reconstruction of the fetal face from 3D US data have been explored previously. Feng *et al*. [8] developed an automatic method for fetal face detection; however, their algorithm results in the creation of a mesh rather than a true surface representation of the face. Bonacina *et al*. [9] describe an automatic method using histogram processing which they applied to five 3D US volumes with good result, although fetal contact with maternal tissue interfered with the algorithmic performance in one of these cases, and Speranza *et al*. detail a technique to extract 3D data from US images which enables 3D models of the fetal face to be printed [10].

In order to overcome the limitations of MS and facilitate the clinical adoption of this technology, we implemented an atlas-based partially automatic segmentation (AS) method, based on algorithms described by Zuluaga *et al*. [11]. Multi-atlas propagation segmentation uses a set of already segmented 'ground-truth' guide images (i.e. 'atlas'), as a reference bank for an algorithm tasked with building the segmentation of a new, unseen image. It selects the most relevant parts of the different atlas images and fuses them in order to create the new image. Preliminary results based on an atlas of 20 images have demonstrated its feasibility for the extraction of 3D faces from fetal 3D US [12]. However, in order to further improve this technique and to enable analysis of the segmented volume, additional refinement of the AS via MS is required (figure 1) in order to clearly define the facial borders, remove extraneous material and complete parts of the face which may be missing (figure 2).

It has been suggested that MS is a subjective process, with variation between operators potentially impairing the accuracy of subsequent quantitative analysis. The variability of MS has been examined previously in other imaging modalities and anatomical structures [13–16]. We identified one study which compared the MS of five utero-fetal unit segmentations to those of an AS method [17]. However, to the best of our knowledge no studies have been conducted examining the variability of MS or the use of an automatic atlas-based segmentation algorithm specifically for the fetal face.

## 1.1. Aims

The aims of this study were to:

1. Evaluate the inter- and intra-operator variability of the manual refinements of automatic segmentations in order to assess the accuracy of this method.
2. Test an optimized version of our partially AS algorithm.

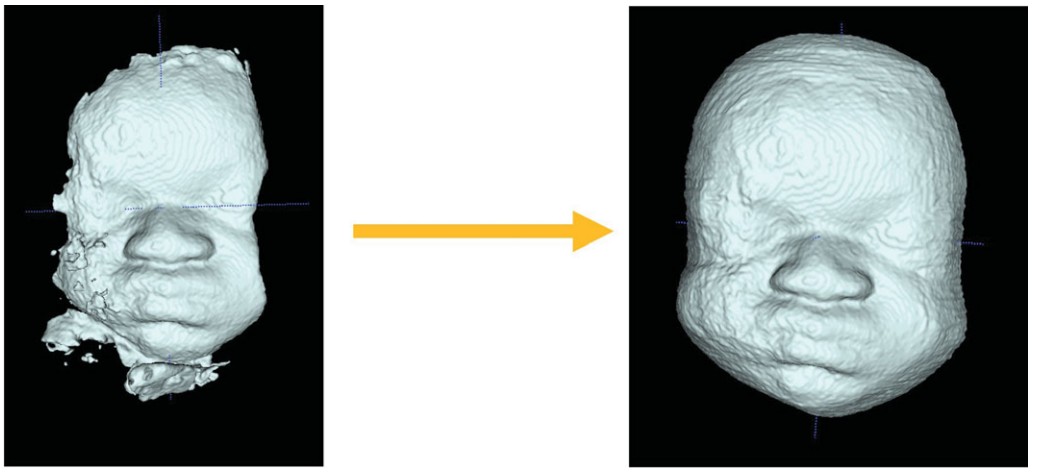

**Figure 1.** Example of an automatically segmented (AS) fetal face before (left) and after (right) manual refinement of the AS has been performed.

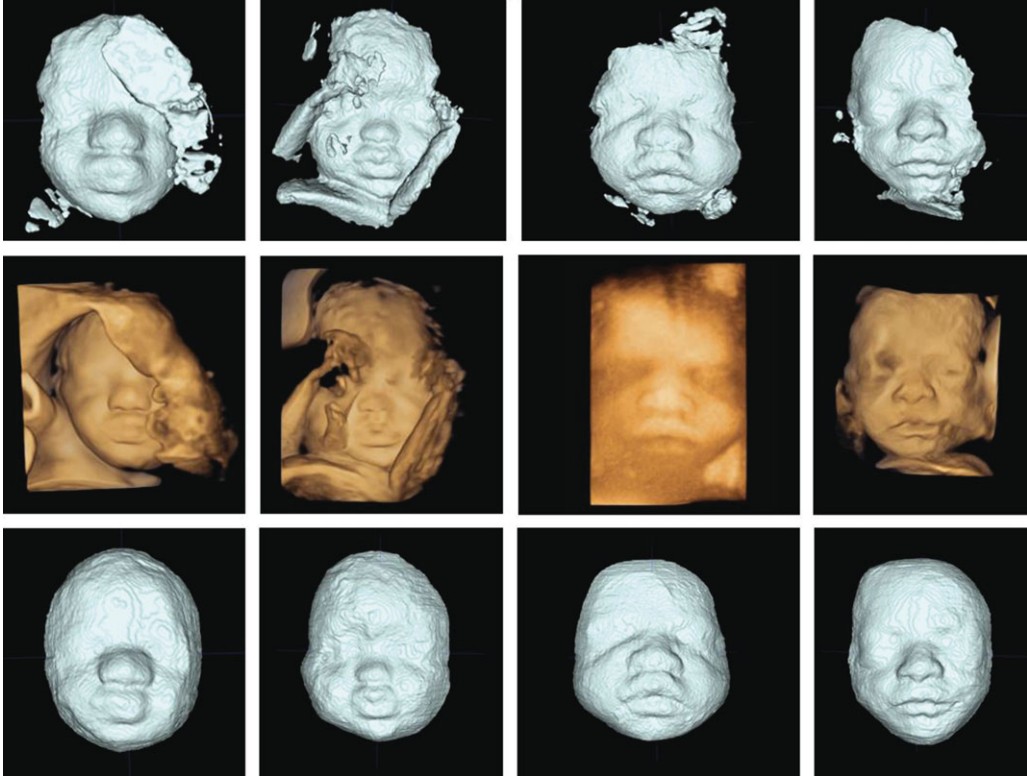

**Figure 2.** Examples of the segmentation produced by the automatic algorithm (top), the corresponding 3D US volume (middle) and the final segmentation following manual refinement (bottom).

## 2. Methods

Thirty-five 3D US volumes of fetal faces above 24 weeks of gestation ($25^{+0}$–$37^{+0}$) were acquired for clinical indications between 2016 and 2018 using Voluson E6, E8 or E10 ultrasound machine (GE Healthcare) with a low-frequency probe (4–8 MHz) and retrospectively included in this study. Cases were defined as normal or abnormal based upon the antenatal US diagnosis, with a total of 11 abnormal and 24 normal cases. The volumes were fully anonymized in order to allow their inclusion in this technical feasibility study. Ethical approval was not required for this study using fully anonymized routinely collected US volumes (Prenatal 3D face study: technical feasibility and

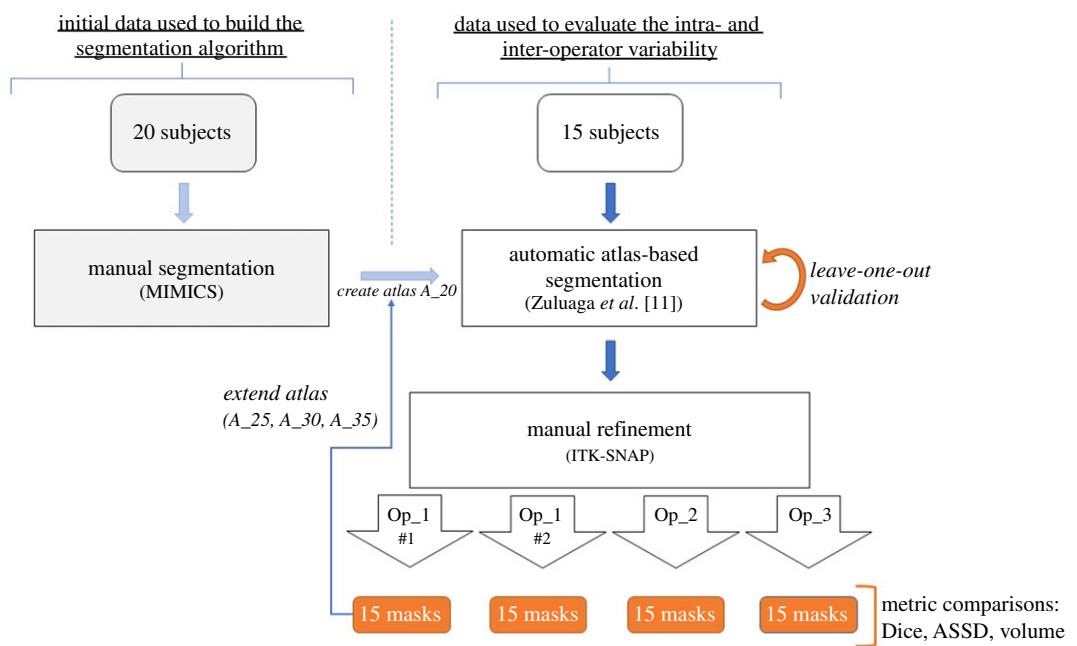

**Figure 3.** A schematic of the workflow for this study. The manual refinements of Op_1#1 were then progressively added to the original set of 20 images in order to increase the atlas size.

improving the methodology) based on information provided to the online Health Research Authority tool (http://www.hra-decisiontools.org.uk/research/). An overview of the main steps of our workflow is illustrated in figure 3.

The 3D volumes were acquired in the midsagittal plane of the fetal face, using the highest resolution acquisition mode and a 3D volume box matched to the size of the fetal face. The volume was acquired where possible without intervening umbilical cord or fetal limbs. The volumes were aligned using multiplanar view along the $x$-, $y$- and $z$-axes and subsequently exported as Cartesian volumes (.vol) for further processing. The first 20 facial volumes were manually segmented in Mimics (Materialise NV, Belgium) and used to create our AS tool [12]. A further 15 facial volumes were segmented via our partially AS tool with additional manual refinement performed in ITK-SNAP (www.itksnap.org) [18]. The mid-points of the posterior border of the widest cross-section of the orbits were used as a landmark to define the depth of manual refinement required in ITK-SNAP.

## 2.1. Inter- and intra-operator manual segmentation assessment

Fifteen additional volumes were initially segmented by our partially AS tool with manual refinement of the automatic segmentations performed on all 15 volumes by three different operators (op_1, op_2 and op_3). Op_1 repeated the segmentation of the 15 volumes twice within a 6-month period denoted by op_1#1 and op_1#2. All operators were blind to one another's manual segmentations.

In order to assess inter-operator variability, a comparison of 15 manual segmentations (MS) was performed between the three operators and intra-operator variability assessed by comparing the 15 repeated segmentations of op_1 (op_1#1-op_2, op_1#1-op_3, op_2-op_3 and op_1#1-op_1#2).

We assessed three commonly used parameters which evaluate overlap, surface distance and segmentation volume in order to assess both the accuracy and reproducibility of the manual segmentations.

## 2.2. Overlap assessment

The degree of overlap between the MS performed by all three operators as well as between the repeated segmentations of op_1 was calculated using the Dice coefficient (0–1). A good overlap is considered when Dice score $\geq 0.7$ [19].

The Dice score [20] was computed as

$$\text{Dice}(M_1, M_2) = 2\frac{|M_1 \cap M_2|}{|M_1| + |M_2|},$$

where $M_1$ and $M_2$ are the segmentation masks and $|M_1|$ and $|M_2|$ are the volumes of these segmented regions. Given the complexity of the fetal face and how subtle the changes seen in dysmorphic facial features can be, we considered a Dice score $\geq 0.8$ to be acceptable in this context.

## 2.3. Surface assessment

The average symmetric surface distance (ASSD) calculates how much the surface varies between two segmentations. The Euclidean distance is calculated between each surface voxel of one segmentation to the closest surface voxel of the other segmentation. The average of all of these distances is then calculated in millimetres, with a result of 0 equivalent to two identical segmentations. The ASSD was calculated as:

$$\mathrm{ASSD}(S_1,S_2) = \frac{1}{2}(\mathrm{mean}_{x \in S_1}\mathrm{dist}(x,S_2) + \mathrm{mean}_{x \in S_2}\mathrm{dist}(x,S_1)),$$

where $S_1$ and $S_2$ are the set of voxels on the surface (in our case the face) of each segmentation, and the distance of a point to a surface is defined by the minimal Euclidean distance from this point to a point on the surface:

$$\mathrm{dist}(x,S) = \mathrm{min}_{y \in S} \parallel x - y \parallel .$$

## 2.4. Volume assessment

The volume of each MS was calculated as the sum of the total segmented voxels. The volumes were then compared by computing the pairwise percentage of volume differences as:

$$\mathrm{diffVol}_{ij} = 2\frac{|V_i - V_j|}{V_i + V_j},$$

where $V_i$ and $V_j$ are the segmentation volumes of the same image as produced by operators $i$ and $j$, respectively.

## 2.5. Automatic segmentation algorithm

A partially automatic atlas-based segmentation (AS) algorithm was implemented, as described by Zuluaga *et al.* [11], that can build a segmentation of a previously unseen image based on a set of already segmented (i.e. 'ground-truth') images or 'atlas'. Default parameters were used for both the non-rigid registration using the *Fast Free-Form Deformation* (F3D) algorithm [21] and the label fusion step using the niftySeg implementation of the *Similarity and Truth Estimation for Propagated Segmentations* (STEPS) algorithm [22]. To understand the optimum number of ground-truth images required for the atlas to achieve the highest quality segmentation in a reasonable computational time, we evaluated the algorithms' performance for different numbers of manually segmented ground-truth images: 20, 25, 30 and 35 images. To do so, we created a growing sequence of atlases (respectively named A_20, A_25, A_30 and A_35) by progressively adding the new manually refined segmentations by op_1#1 to the original set of 20 images. The algorithm was validated by applying a 'leave-one-out' cross-validation scheme on the first 20 images of each atlas, i.e. each of the images was automatically segmented with the remaining (19 for A_20, 24 for A_25… etc.) adopted as atlas. The segmentations resulting from the automatic process were compared with the ground-truth segmentations in terms of visual assessment, Dice score and computational time, to evaluate the tool performance and select the best atlas combination.

## 2.6. Statistical analysis

Statistical analysis was performed using Scipy statistical module [23]; where applicable, individual $p$-values are presented.

In order to examine the inter- and intra-operator variability, a Friedman test was performed for the four pairwise comparisons (op_1#1-op_2, op_1#1-op_3, op_2-op_3 and op_1#1-op_1#2) to assess the consistency of Dice score, ASSD and segmentation volume for all 15 manually refined 3D US volumes between operators (significance level $p < 0.05$). When the null hypothesis was rejected, a post hoc

**Table 1.** Values represent median Dice scores, average symmetrical surface distance (ASSD) and percentage of segmented volume difference of the manual segmentations performed by the three different operators and between the initial and repeated MS performed by op_1. Data are expressed as median $\pm$ IQR (min–max).

| | inter-operator | | | intra-operator |
|---|---|---|---|---|
| | Op_1$_{\#1}$-Op_2 | Op_1$_{\#1}$-Op_3 | Op_2-Op_3 | Op_1$_{\#1}$-Op_1$_{\#2}$ |
| Dice | 0.92 $\pm$ 0.02 | 0.93 $\pm$ 0.03 | 0.90 $\pm$ 0.05 | 0.94 $\pm$ 0.03 |
| | (0.89–0.97) | (0.88–0.97) | (0.87–0.97) | (0.86–0.99) |
| average symmetric surface distance (mm) | 0.44 $\pm$ 0.23 | 0.45 $\pm$ 0.25 | 0.56 $\pm$ 0.32 | 0.33 $\pm$ 0.18 |
| | (0.3–0.79) | (0.22–0.89) | (0.16–1.1) | (0.17–0.7) |
| difference in segmentation volume (%) | 4.8 $\pm$ 4 | 3.5 $\pm$ 3.3 | 2.7 $\pm$ 2.3 | 5.4 $\pm$ 4.5 |
| | (0.0006–11) | (0.7–10.7) | (0.4–9.2) | (0.5–13.3) |

analysis with Bonferoni correction (significance level $p < 0.017$) was performed to identify whether the difference was in the inter- or intra-operator measurements.

Intraclass correlation coefficient (ICC) using repeated measures ANOVA was calculated to compare the total segmentation volumes of each operator.

$T$-test for the linear model between average Dice score, ASSD and the cubic root of the segmentation volume was performed to assess correlation between segmentation size (volume) and inter-operator differences. When assessing the AS algorithm, analysis of variance was performed with a non-parametric Kruskal–Wallis ($p = 0.05$) followed by post hoc testing with Bonferroni correction (significance level $p = 0.0083$) in order to assess difference between Dice score distributions.

# 3. Results

## 3.1. Manual segmentation assessment

Median Dice scores, ASSD and pairwise percentage of volume differences between operators are represented in table 1 and figure 4. When comparing the overall similarity parameters for the four pairwise comparisons (op_1$_{\#1}$-op_2, op_1$_{\#1}$-op_3, op_2-op_3 and op_1$_{\#1}$-op_1$_{\#2}$), a difference was found in Dice scores (Friedman $p = 0.02$) and ASSD ($p < 0.001$) but not in the percentage of segmentation volume difference ($p = 0.16$). While a difference in Dice score between operators was found, the median values for this parameter were all above the 0.7 acceptable minimum threshold, the minimum values in each comparison also exceeding 0.8 which we chose as our internal threshold. Moreover, the central facial region (superimposed for difference operators in figure 4) is the most important for feature identification and also has the best agreement even where intra- and inter-operator Dice score are lowest.

Post hoc analysis demonstrated that these differences in Dice score ($p = 0.03$) and ASSD ($p = 0.002$) can both be attributed to a lower intra-operator variability when compared to the three corresponding inter-operator measurements (table 1). However, this difference is minimal with the differences equal to 0.86 standard deviations for the Dice score and 0.83 standard deviations for the ASSD, and an ICC of 0.98 confirms good overall agreement between operators on the segmentation volume.

There was no correlation found between overall segmentation volume and inter-operator differences for the average Dice score ($T$-test $p = 0.66$) or ASSD ($p = 0.25$) values.

The average time required to perform the MS was 5 h3 m per face (range 3 h2 m–6h 10 m).

## 3.2. Automatic segmentation

The AS tool returned a segmentation for all images. Dice scores for the four atlases tested are plotted in figure 5$a$ for all 20 patients. The boxplot of Dice score distribution is represented in figure 5$b$ and table 2.

Atlas size had an impact on the Dice score (Kruskal–Wallis $p < 0.001$). Post hoc testing demonstrated differences between A_20 versus A_30 ($p = 0.001$), A_20 versus A_35 ($p = 0.001$) and A_25 versus

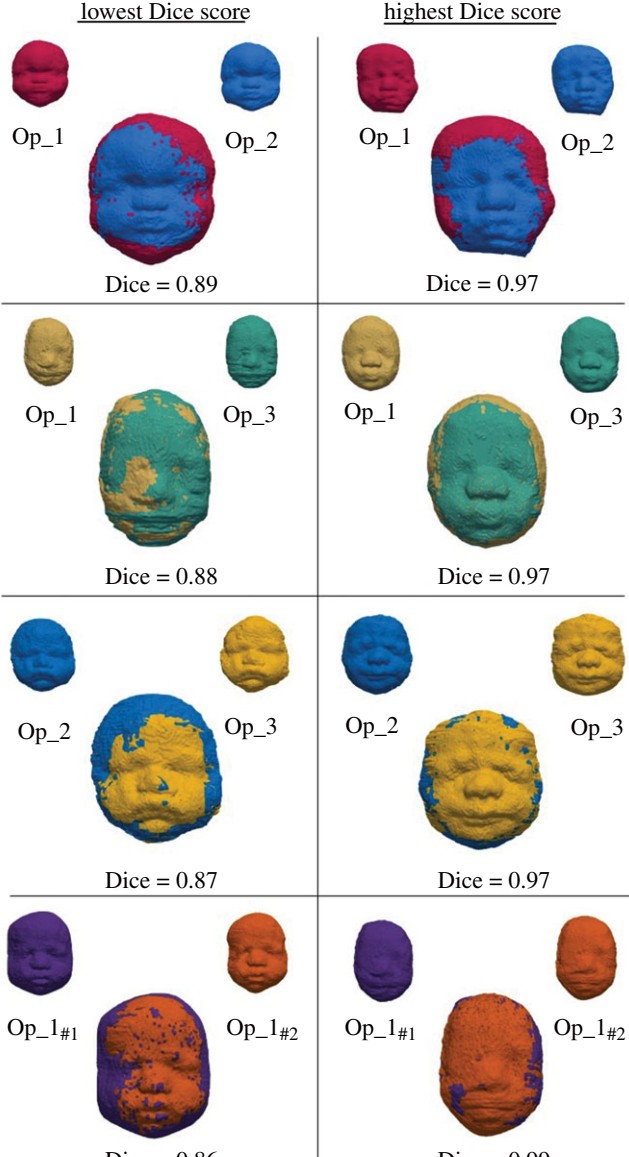

**Figure 4.** Rendering of the lowest (left column) and highest (right column) MS agreement between the three operators and repeated segmentation of operator 1 as assessed by the Dice score.

A_30 ($p = 0.007$) but no difference for A_20 versus A_25 ($p = 0.32$) or A_30 versus A_35 ($p = 0.96$) (table 2 and figure 5*a,b*). The lowest and highest results according to Dice score are illustrated in figure 6 for each atlas, showing that the main facial features could be represented even in the cases of the lowest Dice score.

Segmentations were also compared in terms of volume (figure 5*c,d*). An effect of the number of ground-truth cases on the resulting segmentation volume was found ($p < 0.001$). In the post hoc analysis, differences were found between A_20 and A_30 ($p = 0.001$), A_20 and A_35 ($p < 0.001$), A_25 and A_35 ($p = 0.001$), and to a lesser extent between A_25 and A_30 ($p = 0.005$). There was no difference between A_20 and A_25 ($p = 0.34$) or A_30 and A_35 ($p = 0.42$). The average volume size of the resulting segmentation volume increases with the number of ground-truth images in the atlas but for every atlas (A_20, A_25, A_30 and A_35), the segmented volume is on average smaller than the ground-truth volume ($p < 0.001$).

The computational time is illustrated in figure 5*e,f*, and in table 3. As expected, average computational time increases by increasing the number of ground-truth images in the atlas, with differences ($p < 0.001$) between all comparisons (A_20 versus A_25, A_20 versus A_30, A_20 versus A_35, A_25 versus A_30, A_25 versus A_35 and A_30 versus A_35).

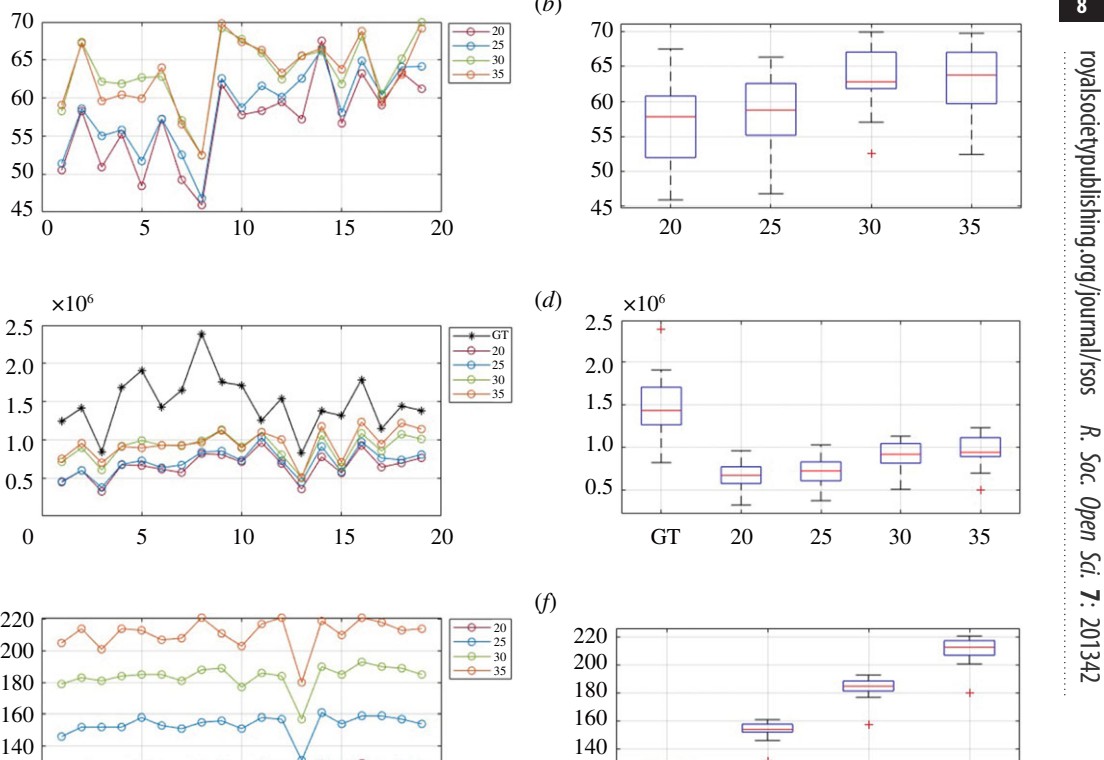

**Figure 5.** Dice score (*a*), segmentation volume (*c*) and computational time (*e*) plotted for each AS, and boxplot of their distribution (*b,d,f*) obtained with the different atlases tested.

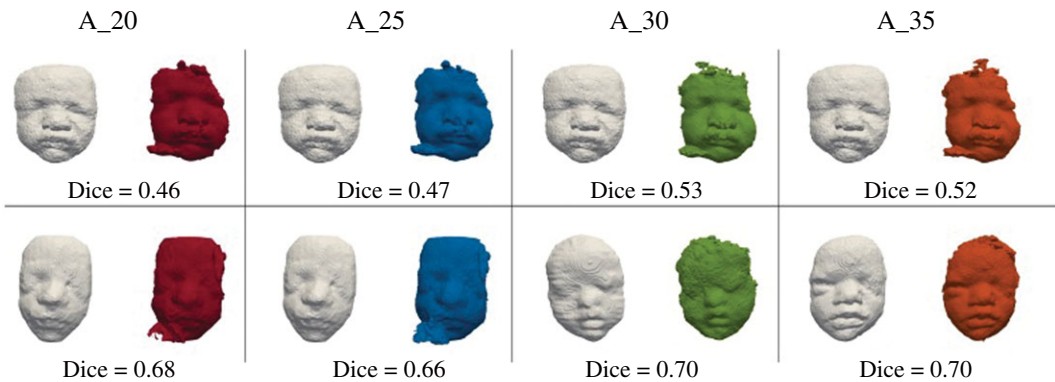

|  | A_20 | A_25 | A_30 | A_35 |
|---|---|---|---|---|
|  | Dice = 0.46 | Dice = 0.47 | Dice = 0.53 | Dice = 0.52 |
|  | Dice = 0.68 | Dice = 0.66 | Dice = 0.70 | Dice = 0.70 |

**Figure 6.** Lowest (top) and highest (bottom) Dice scores for the AS of each atlas: for A_20 (red), A_25 (blue), A_30 (green) and A_35 (orange) compared to the MS (white) as assessed by the Dice score.

**Table 2.** Values represent median Dice scores when comparing the manual segmentations with the automatic segmentations obtained with each of the four different atlases tested. Data is expressed as median ± IQR (min–max).

|  | A_20 | A_25 | A_30 | A_35 |
|---|---|---|---|---|
| Dice | 0.58 ± 0.07 (0.46–0.68) | 0.59 ± 0.07 (0.47–0.66) | 0.63 ± 0.05 (0.53–0.70) | 0.64 ± 0.07 (0.52–0.70) |

Of the four atlases tested, A_30 was the optimum atlas size with an overall improvement in performance balanced by no excessive increase in computational time. When adopting A_35, the computational time significantly increased without a corresponding improvement in performance compared to A_30.

**Table 3.** Values represent median computational time (in minutes) to obtain the AS for each of the four different atlases tested. Data are expressed as median ± IQR (min–max).

| | A_20 | A_25 | A_30 | A_35 |
|---|---|---|---|---|
| time (min) | 125 ± 4 (107–129) | 154 ± 6 (131–161) | 185 ± 7 (157–193) | 213 ± 10 (180–221) |

## 4. Discussion

The aims of this study were to assess the inter- and intra-operator variability of manual refinement of partial AS of the fetal face and to test an optimized version of our AS algorithm. We have shown good reproducibility and accuracy of manual segmentations of fetal face 3D US volumes between operators as well as demonstrating low intra-operator variability with all Dice scores exceeding our internally set threshold of 0.8. While a difference in ASSD was demonstrated, the 'real-life' difference was minimal, equating to a median of 0.44 mm (0.16–1.1 mm). This difference is unlikely to be clinically significant and not large enough to negatively impact upon the accuracy of subsequent diagnostic analysis of the segmentation volumes. We have also shown that it is possible to improve the performance of our AS algorithm by increasing the atlas size which in turn reduces the amount of manual refinement required. However, beyond a certain point, atlas size improvement was minimal and outweighed by significantly increased computational time.

A fully automatic and objective method to identify and characterize abnormal fetal face morphology *in utero* has the potential to both increase prenatal diagnosis of genetic syndromes and as a result improve patient counselling, allow robust delivery and postnatal management planning and provide additional long-term predictive information for parents. Some of these conditions cannot yet be diagnosed through prenatal invasive or non-invasive testing, and others can be, but parents may opt not to undergo testing [24]. Many genetic syndromes and conditions have very distinct craniofacial characteristics. Identifying these abnormalities on prenatal US is largely subjective, requiring significant US operator expertise and there is a propensity to recognize dysmorphic facial features when they coexist with other anatomical abnormalities.

Trisomies are the most identifiable aneuploidies: trisomy 21 fetuses have facial features of brachycephaly, midface hypoplasia, flattened nasal bridge and macroglossia, while trisomy 18 features a prominent occiput, low-set ears, and micrognathia [25], however, these conditions are readily diagnosable from invasive and non-invasive prenatal diagnosis. By contrast, rarer syndromes cannot always be diagnosed from prenatal testing as in many the genetic abnormality is not known and in some a strong clinical suspicion of a diagnosis or differential diagnoses is required for targeted genetic testing [24,26,27]. These may also be associated with developmental or physical impairments as the child grows older, hence the importance of prenatal diagnosis [28–30].

Such conditions include Crouzon syndrome which is characterised by brachycephaly, frontal bossing, shallow orbits and maxilla hypoplasia, and Apert syndrome with flat facies, prominent forehead and hypertelorism [25,30]. Frontal bossing, a flattened nasal bridge and micrognathia is a feature of many skeletal dysplasias and although severity can vary significantly, the malar hypoplasia, zygomatic bone cleft and down-slanting palpebral fissures can be very distinct in Treacher Collins syndrome [25]. The diagnosis of Cornelia de Lange syndrome (CdLS) is rarely made on prenatal US but has profound lifelong implications. Even when CdLS is suspected prenatally the diagnosis can remain elusive as 30% of cases have unknown genetic aetiology [29]. The diagnosis is most commonly made clinically in the neonatal and paediatric period with distinct facial characteristics including low-set ears, short upturned nose, synophris, everted nostrils, micrognathia and long philtrum giving the diagnostic clues [31].

This study has some limitations. The number of segmentations performed was modest, although sufficient to demonstrate the low variability between segmentations and acceptability of this method. The operators were of differing skill levels, with two operators having no prior experience of performing or interpreting US images. This had the potential to negatively impact on the variability between manual segmentations; however, our results suggest that this was not the case and rather, reinforce the reliability of the manual refinement. There was also a potential for the operators to become more competent in manual segmentations over time, which may have impacted on the variability by reducing it. This was not the case: two of the operators performed the manual segmentations in the same order, with no differences between their performance and that of the other operator demonstrated.

Performing segmentations of US volumes, in comparison to other imaging modalities such as CT and MRI, is challenging given the variations in US image quality through acquisition, operator skill, movement artefact, maternal body habitus, amniotic fluid volume and fetal position. These all result in reduced image quality and, therefore, reduced boundary definition within parts of the US image. This then has the potential to cause higher variability between manual segmentations. However, our results suggest that these did not particularly affect the study and, given that US remains the imaging modality of choice in anomaly screening and in specialist fetal medicine units, are reassuring in regard to the potential application of this method in clinical practice.

The main limitation of this technique and barrier to use in clinical practice is the need for additional manual refinement and the time taken to perform this. The median segmentation time in this study was approximately 5 h per fetal facial volume, which is clearly impractical in everyday clinical practice and would preclude this technique from clinical use in its current form. With any segmentation method used to reconstruct the fetal face, as noted previously, the main concern would be in ensuring accuracy and avoiding the normalizing of an abnormal face or vice versa. At the present time, state-of-art methods are unable to accurately segment fetal faces from 3D US images while being robust to a highly variable context (the fetal hand or umbilical cord in contact with or obstructing the face, contact with the uterine wall or placenta) and poor image quality [9]. These limitations are partially overcome by our automatic algorithm with promising results despite the inability to produce a complete segmentation of the face in the most difficult cases resulting in the need for manual corrections. We have demonstrated that the consistency of the manual segmentations is a good foundation for atlas-based segmentation algorithms. Moreover, the improvements seen in our partially automatic algorithm performance in regard to improved Dice scores and therefore accuracy associated with an increased number of reference images (atlas) are positive. Further work will focus on more accurate and faster registration algorithms in order to move closer to a fully automated method suitable for clinical 'point of diagnosis' use.

## 5. Conclusion

We have demonstrated the technical feasibility of fetal facial segmentation using manual refinement of partially automatically segmented US facial volumes. Moreover, the quality of AS can be improved by increasing the atlas size and training the algorithm with a larger number of manually segmented volumes thus reducing the need for manual refinement of the automatic segmentations. Though not yet feasible, given the improvements that we have demonstrated in these techniques, it is possible to consider that further refinement would lead to a fully automated method for the reconstruction and quantification of the fetal face. This is a prerequisite for clinical utility, as to be an adjunct to prenatal diagnosis from fetal facial morphological appearance such a system would need to work almost in real time with the US scan itself.

Data accessibility. The dataset used in this study consists of fully anonymized US images. As outlined in our methods, ethical approval was not required for this technical feasibility study as it did not fulfil the HRA requirements for research ethics consideration. Therefore, patients have not consented for their US volumes to be publicly available. With the agreement of the journal's Editorial Office, the code for the fetal face segmentation tool used in this paper is openly available on R.S.'s GitHub page: https://github.com/rsivera/fetal-faces-us_segmentation.
Authors' contributions. C.C.L., S.S., A.E.C., B.B. and A.D. conceived and designed the study; A.E.C., A.D., C.C.L., L.F., T.-L.W. and G.P. were involved in data collection; A.E.C., B.B. and R.S. analysed the data for publication. A.C., B.B., R.S., A.D., S.S., L.F., T.-L.W., G.P., D.D. and C.C.L. drafted the article and revised it for important intellectual content.
Competing interests. The authors declare no competing interests, financial or otherwise.
Funding. This work was supported by the National Institute for Health Research (NIHR) Biomedical Research Centre based at Imperial College Healthcare NHS Trust and Imperial College London (grant no. P74419). The views expressed are those of the author(s) and not necessarily those of the NHS, the NIHR or the Department of Health. This work was also supported by the Engineering and Physical Science Research Council (grant no. EP/N02124X/1) and the European Research Council (grant no. ERC-2017-StG-757923).

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
