## [Reviewer comments · Royal Society Open Science]

Review History

RSOS-201342.R0 (Original submission)

Review form: Reviewer 1 (Giuseppe Rizzo)

Is the manuscript scientifically sound in its present form?

Yes

Are the interpretations and conclusions justified by the results?

Yes

Is the language acceptable?

Yes

Do you have any ethical concerns with this paper?

No

Have you any concerns about statistical analyses in this paper?

No

Recommendation?

Accept as is

Comments to the Author(s)

Congratulations nicely reviewed

Review form: Reviewer 2

Is the manuscript scientifically sound in its present form?

Yes

Are the interpretations and conclusions justified by the results?

Yes

Is the language acceptable?

Yes

Do you have any ethical concerns with this paper?

No

Have you any concerns about statistical analyses in this paper?

No

Recommendation?

Accept as is

Comments to the Author(s)

Authors have improved the scientific level of the paper I support the paper publication

Decision letter (RSOS-201342.R0)

Dear Dr Clark:

It is a pleasure to accept your manuscript entitled "Developing and testing an algorithm for automatic segmentation of the fetal face from 3D ultrasound images" in its current form for publication in Royal Society Open Science. The comments of the reviewer(s) who reviewed your manuscript are included at the foot of this letter.

Please also ensure that your code is deposited in a suitable repository, per the guidance at <https://guides.github.com/activities/citable-code/>. When you have deposited the code, you should ensure that an updated reference is included in your paper's bibliography.

on behalf of Dr Francois Fages (Associate Editor) and Professor Marta Kwiatkowska (Subject Editor).

Associate Editor Dr Francois Fages Comments to Author:

Associate Editor: 1
Comments to the Author:
Dear authors

It is my pleasure to accept your revised version as is, since it seems to have satisfactorily answered the previous criticisms of the reviewers.
Best regards

Reviewer(s)' Comments to Author:
Reviewer: 1

Comments to the Author(s)
Congratulations nicely reviewed

Reviewer: 2

Comments to the Author(s)
Authors have improved the scientific level of the paper I support the paper publication
